# Selective cobalt and nickel electrodeposition for lithium-ion battery recycling through integrated electrolyte and interface control

Kwiyong Kim[1], Darien Raymond [1], Riccardo Candeago [1] & Xiao Su [1✉]

Molecularly-selective metal separations are key to sustainable recycling of Li-ion battery electrodes. However, metals with close reduction potentials present a fundamental challenge for selective electrodeposition, especially for critical elements such as cobalt and nickel. Here, we demonstrate the synergistic combination of electrolyte control and interfacial design to achieve molecular selectivity for cobalt and nickel during potential-dependent electrodeposition. Concentrated chloride allows for the speciation control via distinct formation of anionic cobalt chloride complex ($CoCl_4^{2-}$), while maintaining nickel in the cationic form ($[Ni(H_2O)_5Cl]^+$). Furthermore, functionalizing electrodes with a positively charged polyelectrolyte (i.e., poly(diallyldimethylammonium) chloride) changes the mobility of $CoCl_4^{2-}$ by electrostatic stabilization, which tunes cobalt selectivity depending on the polyelectrolyte loading. This strategy is applied for the multicomponent metal recovery from commercially-sourced lithium nickel manganese cobalt oxide electrodes. We report a final purity of $96.4 \pm 3.1\%$ and $94.1 \pm 2.3\%$ for cobalt and nickel, respectively. Based on a technoeconomic analysis, we identify the limiting costs arising from the background electrolyte, and provide a promising outlook of selective electrodeposition as an efficient separation approach for battery recycling.

[1] Department of Chemical and Biomolecular Engineering, University of Illinois at Urbana-Champaign, Urbana, IL 61801, USA. ✉email: x2su@illinois.edu

Worldwide consumption of electronic devices has led to a sharp increase in waste batteries[1]. Spent lithium-ion batteries (LIBs) contain critical elements, such as lithium (5–8%), cobalt (5–20%), nickel (5–10%), and manganese (10–15%), and nickel–metal hydride batteries also possess a high content of nickel (36–42%) and cobalt (3–5%)[2,3]. The future demand for critical elements, especially cobalt and nickel, has been predicted to exceed identified reserves[1,4–6], and there are increasing geographical, environmental, and political pressures related to primary mining operations[7,8]. Thus, there is urgent pressure to develop sustainable strategies to recover critical elements from the potentially valuable secondary resources[9].

Considering the high content of valuable *d*-block elements, the recycling of multi-metallic cathodes, such as lithium nickel manganese cobalt oxide (NMC) cathode, has received particular attention. In general, hydrometallurgical processes for cathode recycling involve a series of pretreatment steps, including discharging, dismantling, separating, and harvesting of active materials from a current collector[10]. In a subsequent leaching step, the constituent elements in the solid phase are transferred into a liquid phase for further purification. The selective separation of cobalt and nickel from post-leaching solution is critical to ensuring a sustainable method of recovering each constituent metal with high purity, but it is challenging due to the similar physicochemical properties of cobalt and nickel. State-of-the-art recycling processes (e.g., LithoRec process, a laboratory-scale process by Aalto University) rely on solvent extraction, precipitation, or a combination of these as a way of separation of cobalt and nickel[11]. Also, there have been extensive studies at a laboratory scale for the separation of cobalt and nickel, such as solvent extraction[12,13], precipitation[14], adsorption[15–21], intercalation[22], and dialysis[23], all of which can be beneficial for cobalt/nickel recovery in the NMC chemistry regime. A comparison of different state-of-the-art cobalt/nickel separation techniques is summarized in Supplementary Table 1 to provide benchmarks for selectivity. Of particular note, solvent extraction and precipitation usually exhibit high selectivity performance but can often incur large chemical costs or waste and may face challenges concerning complex solution/speciation chemistry[9,24]. As such, technologies that can complement or assist in process intensification of these complex purification trains are urgently needed, especially if they can lower either thermal/chemical consumption or waste generation.

As an alternative, electrochemical methods have been suggested as a promising approach, which, in combination with renewable sources, allow for sustainable and distributed processes for metal recycling[25]. Among various electrochemically driven techniques, electrodeposition is a versatile and simple method with tunability in nucleation and growth, morphology, and deposit composition[26–28]. Electrodeposition has been useful for the separation and recovery of metals from multicomponent mixtures[29–31], with the reduction potentials of component metals being the most crucial parameter dictating selectivity[25]. Reduction potentials of cobalt and nickel have been systematically investigated in high temperature eutectic mixtures of molten LiCl–KCl and NaCl–KCl[32,33]. These early studies have provided the basis of selective electrochemical deposition of cobalt and nickel in fused salt[34,35], but high operational temperatures (400–550 °C) are not desirable considering the problematic integration with hydrometallurgical processes. Low-temperature, aqueous-based alternatives are thus a more desirable, environmentally compatible, and energy-efficient route. However, aqueous electrolytes present intrinsic selectivity limitations for electrodeposition processes, due to the similar standard reduction potentials between cobalt and nickel ($E°_{Co} = -0.277$ V and $E°_{Ni} = -0.250$ V vs standard hydrogen electrode (SHE))[36],

leading to unwanted co-deposition with low selectivity[31] and thus necessitating additional chemical steps for cobalt/nickel separation (e.g., solvent extraction) before recovery via electrowinning[10]. Therefore, it is necessary to find innovative ways that can control selectivity between cobalt and nickel, which enable direct separative recovery in aqueous streams with minimized energy input and chemical footprint.

In this work, we highlight fundamental and unique insights into selectivity tuning during electrodeposition at a polymer interface. While recent studies have explored stabilization and morphology control of tailored polymers for lithium electrodeposition[37,38], there has been a lack of comprehensive studies that leverage polymer interfaces for splitting potentials of metal deposition and thus achieve selective metal recovery by differential electrodeposition. Here, by taking advantage of the synergistic combination of electrolyte control and interfacial design, we demonstrate an electrochemical approach for tuning molecular selectivity in cobalt and nickel recovery.

First, we prove that the control of speciation provides an effective electrolyte engineering approach to discriminate metals with similar electrochemical properties in aqueous solutions. Second, we explore interfacial tailoring of the electrode with a positively charged polyelectrolyte, poly(diallyldimethylammonium chloride) (PDADMA), for additional selectivity control enabled by modulating the mobility of $CoCl_4^{2-}$ in the positive polyelectrolyte layer. Through systematic separation tests, electrochemical characterization, spectroscopic, and in situ electrogravimetric analysis, we elucidate the synergistic contributions from electrolyte and interface engineering for tunable selectivity in electrodeposition of cobalt and nickel. Our findings suggest that metal selectivity depends on electrode potential and polymer loading (Fig. 1), thus leading to a surface-tunable method for direct separation of cobalt and nickel in aqueous solutions. Furthermore, we report an electrochemical route for the separative recovery of cobalt and nickel in spent NMC cathodes, enabled by electrolyte- and polymer-driven splitting of reduction potentials and sequential electrodepositions, which do not rely on the intensive use of specialized extractants.

## Results

**Speciation control of cobalt and nickel.** The close reduction potentials of cobalt and nickel present intrinsic difficulties to selectively electrodeposit one metal over the other[31]. With traditional background electrolytes containing low-to-moderate chloride concentrations (e.g., 0.1 M $Li_2SO_4$ and 0.1 M LiCl), cobalt and nickel exhibited similar patterns in linear sweep voltammetry (LSV) curves, and their onset potentials were not easily distinguishable (Fig. 2a, b), due to the predominant cationic speciation of $[Co(H_2O)_6]^{2+}$ and $[Ni(H_2O)_6]^{2+}$. Our strategy is to control speciation along with an integrated view of leaching and recovery. In hydrometallurgical processes, the recovery step can benefit from a preceding leaching step in relation to solution chemistry and speciation control[25]. Concentrated chloride has been used as a ligand for integrated leaching and recovery (e.g., concentrated LiCl, ionic liquids and deep eutectic solvents for ionometallurgy)[39–43]. In this study, we make use of concentrated chloride (10 M LiCl as a model electrolyte) as a background electrolyte for speciation control, which helps the formation of a stable anionic tetrachloro complex ($CoCl_4^{2-}$)[44]. In this electrolyte, nickel exists as the cation $[Ni(H_2O)_5Cl]^+$ so opposite charges can be imparted[45]. The LSV curves of cobalt and nickel showed a distinguishable difference in the onset potentials ($-0.68$ V vs Ag/AgCl and $-0.59$ V vs Ag/AgCl for cobalt and nickel, respectively; Fig. 2c), indicating a feasible separation window where nickel can be selectively electrodeposited. The negative shift in the cobalt

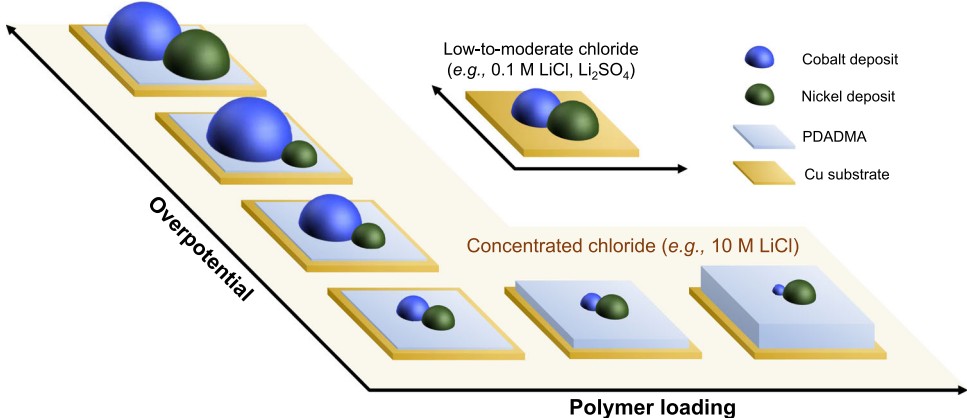

**Fig. 1 A schematic representation of potential-dependent selectivity tuning enabled by synergistic electrolyte and interfacial control.** In concentrated chloride, controlling applied potential and/or polyelectrolyte loading (PDADMA, poly(diallyldimethylammonium chloride)) allows to tune molecular selectivity of cobalt and nickel in electrodeposition processes.

electrodeposition can be attributed to the stabilization by the formation of a stable $CoCl_4^{2-}$ complex[46,47]. During the cathodic LSV sweep toward a more negative range ($<-0.69$ V vs Ag/AgCl), cobalt showed a steeper increase of current magnitude followed by a shoulder ($-0.72$ V vs Ag/AgCl), while the current magnitude for nickel grew slowly near its onset potential and then exhibited gradual enhancement in a more negative range ($<-0.75$ V vs Ag/AgCl; Fig. 2c). The slower growth in the current magnitude during nickel electrodeposition can be ascribed to the sluggish dehydration of the $[Ni(H_2O)_5Cl]^+$ complex[45].

When chronoamperometric electrodeposition tests were carried out in a binary 1:1 mixture of 10 mM Co(II) and Ni(II), the surface Co/Ni ratios were in the range of 1–2 throughout the entire potential range tested ($-0.8$ to $-0.55$ V vs Ag/AgCl) with low-to-moderate chloride concentrations (e.g., 0.1 M $Li_2SO_4$ and 0.1 M LiCl; Fig. 2d, e), indicating difficulty in selectively depositing one specific metal while suppressing the other. On the other hand, in 10 M LiCl, the deposit composition analysis revealed that the moderate applied potentials ($-0.60$ to $-0.55$ V vs Ag/AgCl) promoted the formation of deposits with higher nickel composition (Fig. 2f). Interestingly, the compositions at the relatively negative region ($<-0.65$ V vs Ag/AgCl) revealed the formation of cobalt-selective electrodeposits, which, at $-0.75$ V vs Ag/AgCl, showed the highest Co/Ni ratio of 3.18 (Fig. 2f). Applying more negative potential ($<-0.8$ V vs Ag/AgCl) resulted in Co/Ni ratio close to 1, implying a similar degree of co-deposition of the two metals. The cobalt-selective electrodeposition in the potential range of $-0.65$ to $-0.8$ V vs Ag/AgCl can be ascribed to an effect referred to as anomalous deposition, in which cobalt ($E°_{Co} = -0.277$ V vs SHE) is more preferentially deposited compared to nickel ($E°_{Ni} = -0.250$ V vs SHE)[48]. In this study, the degree of the anomalous deposition in terms of Co/Ni ratio was the highest in 10 M LiCl (at $-0.75$ V vs Ag/AgCl), compared to 0.1 M LiCl and 0.1 M $Li_2SO_4$ (Fig. 2d–f). When the initial concentrations of both metals were increased to 100 mM, the Co/Ni ratio greatly increased even more, reaching values up to 14 (Supplementary Fig. 1). A similar behavior was observed using other cathodic substrates for electrodeposition, too (Supplementary Fig. 2).

**Electrogravimetric analysis of cobalt/nickel anomalous deposition.** To obtain insights into the electrochemical reaction during the electrodeposition, electrochemical quartz crystal microbalance (EQCM) measurements and analyses were carried out. By combining the change in mass with Faraday's law, the specific mass change per the number of electrons could be determined—namely $m/z$ (g mol$^{-1}$)—which is a useful parameter

for studying faradaic processes and quantifying their associated efficiencies[49]. For example, the direct cobalt reduction takes place according to this reaction[50]:

$$Co(II) + 2e^- \rightarrow Co(s) \qquad (1)$$

where the corresponding theoretical $m/z$ value is 29.5 g mol$^{-1}$ (atomic weight of cobalt/2e$^-$ = 58.9 g mol$^{-1}$/2e$^-$). If hydrogen evolution takes place simultaneously, cobalt electrodeposition can also occur through the formation of cobalt hydroxide[50]:

$$2H_2O + 2e^- \rightarrow H_2 + 2OH^- \qquad (2)$$

$$Co(II) + 2OH^- \rightarrow Co(OH)_2(s) \qquad (3)$$

where the corresponding theoretical $m/z$ value is 46.5 g mol$^{-1}$ (molecular weight ($M_W$) of cobalt hydroxide/2e$^-$ = 92.9 g mol$^{-1}$/ 2e$^-$). In the same way, the theoretical $m/z$ value for direct nickel reduction (29.3 g mol$^{-1}$) and nickel hydroxide formation (46.4 g mol$^{-1}$) could be determined.

First, at moderate overpotentials, such as $-0.625$ and $-0.725$ V vs Ag/AgCl for 10 mM Ni(II) and Co(II), respectively, the electrode mass steadily increased due to electrodeposition (Supplementary Fig. 3a, b). It was observed that the $m/z$ value was about only 10 g mol$^{-1}$ in the Ni(II) bath, which was lower than in the Co(II) bath (Supplementary Fig. 3c, d), and this finding is in accordance with relatively low faradaic efficiency of nickel deposition (Supplementary Fig. 4a). On the other hand, in 10 mM Co(II) in 10 M LiCl, higher faradaic efficiencies (>90%) were observed near the onset potentials of cobalt deposition (Supplementary Fig. 4b). Accordingly, the $m/z$ value increased to $51.2 \pm 0.3$ g mol$^{-1}$ in the first 1 min in the Co(II) bath, which is compatible with $Co(OH)_2$ formation according to Eqs. (2) and (3), followed by gradual decrease in $m/z$ ratio (Supplementary Fig. 3d), indicating that $Co(OH)_2$ is formed at the early stage of the electrodeposition because of local pH increase (Eqs. (2) and (3)). The subsequent decrease in $m/z$ ratio can be ascribed to: (1) cobalt deposition via a direct pathway ($m/z = 29.5$ g mol$^{-1}$, Eq. (1)) and/or (2) hydrogen evolution on electrodeposited catalytic cobalt sites, as reported earlier[51]. The formation of $Co(OH)_2$ was also observed at a higher overpotential of $-0.8$ V vs Ag/AgCl in 10 mM Co(II), while Ni(II) still exhibited similar $m/z$ value (~10 g mol$^{-1}$) (Supplementary Fig. 5). The process of $Co(OH)_2$ generation also involves the formation of cobalt monohydroxide as an intermediate:

$$Co(II) + OH^- \rightarrow CoOH^+ \qquad (4)$$

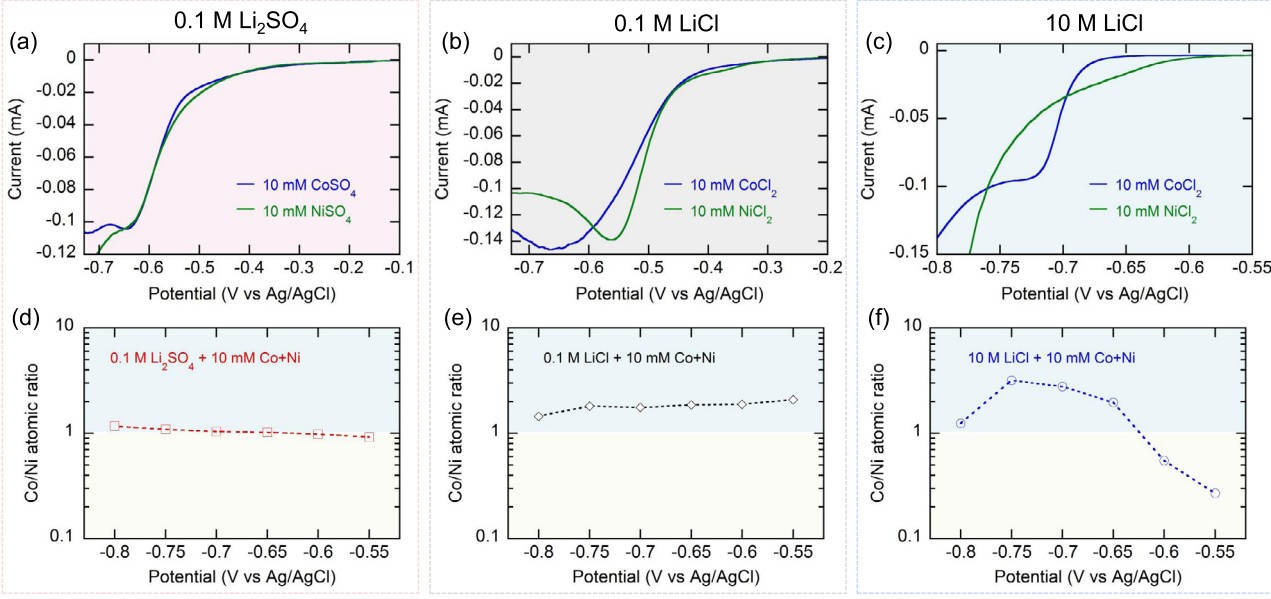

**Fig. 2 Electrolyte control for selective electrodeposition of cobalt and nickel.** Linear sweep voltammograms of a single metal salt of 10 mM Co(II) or Ni(II) in **a** 0.1 M Li$_2$SO$_4$, **b** 0.1 M LiCl, and **c** 10 M LiCl at a scan rate of 5 mV s$^{-1}$. Surface Co/Ni ratios on the electrodeposit formed in the binary mixture of 10 mM Co(II)+Ni(II) in the background electrolyte of **d** 0.1 M Li$_2$SO$_4$, **e** 0.1 M LiCl, and **f** 10 M LiCl.

$$CoOH^+ \rightarrow CoOH^+_{ads} \qquad (5)$$

CoOH$^+$/Co(OH)$_2$ have higher adsorption ability compared to NiOH$^+$/Ni(OH)$_2$[48,52] and thus play a critical role in inhibiting nickel deposition, leading to the anomalous electrodeposition with highly prioritized cobalt deposition[36]. The unique transition from normal to anomalous electrodeposition in concentrated chloride offers an innovative venue of potential-dependent selectivity tuning. Furthermore, concentrated chloride displayed an additional benefit of effective suppression of hydrogen evolution, as reflected in higher faradaic efficiency compared to low-to-moderate chloride electrolyte conditions (Supplementary Fig. 6).

**The effect of PDADMA on tuning electrodeposition selectivity.** In the next step, we prepared functional polymer-coated electrodes and combined with the electrolyte-tuning approach discussed above. Considering the opposite charges of cobalt and nickel, and the pronounced molecular interaction between CoCl$_4^{2-}$ and quaternary amine[53], a positively charged polyelectrolyte, PDADMA ($M_W$ 200,000–350,000 Da) was loaded on the surface of pristine copper foil, and its effect on selectivity was investigated. PDADMA loaded on pristine copper foil exhibited smooth and uniform coating in general (Supplementary Fig. 7), but at relatively high PDADMA loading (e.g., 0.75 mg cm$^{-2}$), unevenly distributed cracks were observed (Supplementary Fig. 8). As shown in Fig. 3a, relatively small loadings (≤0.075 mg cm$^{-2}$) improved the surface Co/Ni ratio compared to a pristine Cu substrate. As shown in Fig. 3b, any PDADMA-loaded surface (PDADMA/Cu) moderated the total amount of metal deposited as compared to pristine Cu because of increased surface resistance. When pristine Cu (0 mg cm$^{-2}$) was compared with a small loading of PDADMA/Cu (0.0375 mg cm$^{-2}$), the degree of PDADMA-driven deposition suppression was larger for nickel as compared to cobalt, because positively charged PDADMA is hypothesized to assist the binding of negatively

charged CoCl$_4^{2-}$. Interestingly, as PDADMA loading is further increased, the amount of cobalt on electrodeposits kept decreasing, while the nickel content was maintained (Fig. 3b). Accordingly, we observed the transition from cobalt-selective to nickel-selective electrodeposition tuned by PDADMA loading; at the potential of −0.725 V vs Ag/AgCl, the surface Co/Ni ratio was 2.3 for pristine Cu, but it decreased to 0.40 for the electrode with PDADMA loading of 4.995 mg cm$^{-2}$ (Fig. 3a).

To elucidate how cobalt electrodeposition is suppressed with the increase in PDADMA loading (>0.0375 mg cm$^{-2}$), a single metal salt of 10 mM Ni(II) or Co(II) was tested with LSV using pristine copper or PDADMA/Cu (0.75 mg cm$^{-2}$; Fig. 3c). In the case of 10 mM Ni(II) in 10 M LiCl, the LSV signal was not significantly affected by the presence of PDADMA on the substrate, exhibiting almost the same onset potential ($|\Delta E_{onset}| = 0.004$ V) to pristine copper. In the case of 10 mM Co(II), however, PDADMA/Cu exhibited a significantly decreased LSV signal with reduced current and a discernable negative shift in the onset potential ($|\Delta E_{onset}| = 0.02$ V), indicating suppressed cobalt electrodeposition and splitting between the reduction potential of cobalt and nickel. When chronoamperometric electrodeposition was carried out at −0.725 V vs Ag/AgCl in the single metal salt solution consisting of 10 mM Ni(II) or 10 mM Co(II) in 10 M LiCl, the amount of electrodeposited nickel was similar between pristine Cu and PDADMA/Cu, while cobalt electrodeposition on PDADMA/Cu accounted only for 7% of pristine Cu (Fig. 3d). We ascribe the inhibiting role of PDADMA in cobalt electrodeposition to the limited mobility of CoCl$_4^{2-}$ localized in the positively charged PDADMA film. To prove this, further LSV analysis was carried out in a single metal salt of either 10 mM Ni(II) or 10 mM Co(II) in 10 M LiCl at various scan rates, with or without 0.01 wt% of PDADMA added as a homogenous additive into the liquid phase (Supplementary Fig. 9). Without PDADMA, 10 mM Co(II) displayed a linear relation between the peak current and the square root of scan rate, suggesting Co(II) reduction is diffusion-controlled (Supplementary Fig. 9a). Surprisingly, adding 0.01 wt% PDADMA significantly suppressed the current (Supplementary Fig. 9b). The calculation of the diffusion coefficient of CoCl$_4^{2-}$ mixed with 0.01 wt% PDADMA showed a dramatic decrease ($4.19 \times 10^{-10}$ cm$^2$ s$^{-1}$) compared to CoCl$_4^{2-}$ in

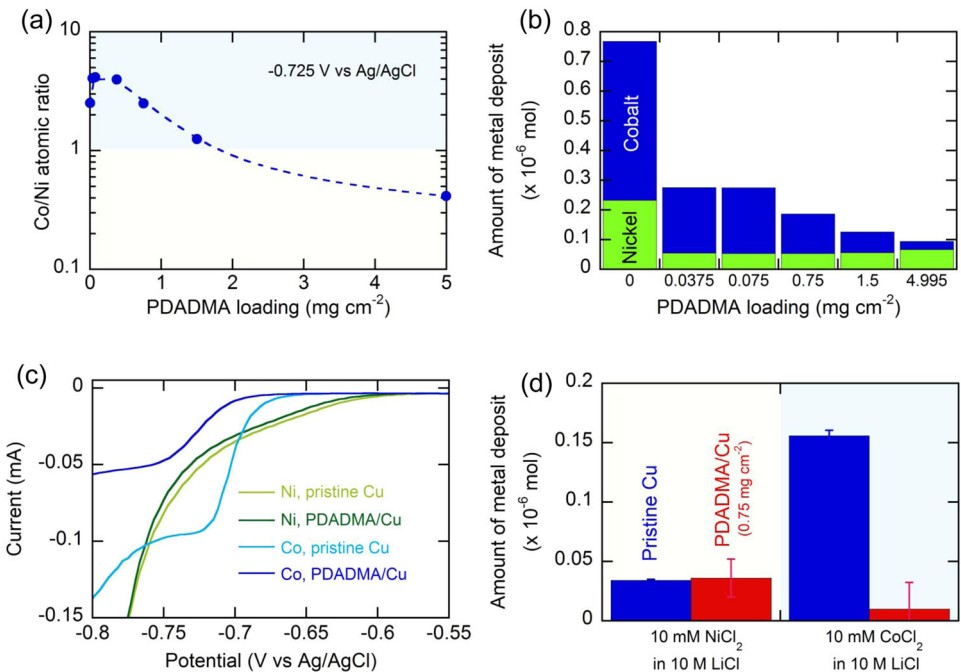

**Fig. 3 Selectivity tuning in the electrodeposition of cobalt and nickel enabled by interfacial charge control with a positive polyelectrolyte, PDADMA.** The effect of PDADMA loading on **a** the surface Co/Ni ratio on the electrodeposit and **b** the actual amount of cobalt and nickel electrodeposited. −0.725 V vs Ag/AgCl was applied for 0.5 h. **c** Linear sweep voltammograms of a single metal salt of 10 mM Co(II) or Ni(II) in 10 M LiCl using pristine Cu and PDADMA-loaded Cu (PDADMA loading: 0.75 mg cm$^{-2}$) at a scan rate of 5 mV s$^{-1}$. **d** The actual amount of cobalt and nickel electrodeposited in a single metal salt of 10 mM Co(II) or Ni(II) in 10 M LiCl using pristine Cu and PDADMA-loaded Cu (PDADMA loading: 0.75 mg cm$^{-2}$). −0.725 V vs Ag/AgCl was applied for 0.5 h in **d**. Error bars indicate standard error of the mean ($n = 3$).

the bulk electrolyte without the polyelectrolyte ($2.50 \times 10^{-8}$ cm$^2$ s$^{-1}$), indicating decreased mobility of CoCl$_4^{2-}$ due to the stabilization effect of PDADMA.

On the other hand, the diffusion coefficient of 10 mM Ni(II) was $1.56 \times 10^{-8}$ and $1.43 \times 10^{-8}$ cm$^2$ s$^{-1}$ without and with the addition of 0.01 wt% PDADMA, respectively, suggesting that the mobility of Ni(II) in 10 M LiCl was not significantly affected by the presence of PDADMA, in contrast to Co(II) (Supplementary Fig. 9c, d). The distinct sensitivity of cobalt and nickel to PDADMA was also reflected in the Tafel plot (Supplementary Fig. 10). The PDADMA coating caused an increase in Tafel slope for 10 mM Co(II) (from 47 to 67 mV dec$^{-1}$), but only a small change was observed for 10 mM Ni(II) (from 149 mV dec$^{-1}$ for pristine Cu to 147 mV dec$^{-1}$ for PDADMA/Cu). These results shed insight into the discrimination in molecular interactions toward different metal ions by the polymer layers and its effect on the selectivity during electrodeposition.

**Synergistic electrolyte and interfacial control for optimized electrodeposition selectivity.** The above results showed that modulation of surface charge allows for selectivity tuning in the metal separation, by enhancing the cobalt selectivity with a low polymer loading and the suppression of cobalt deposition with high polymer loading. At −0.6 V vs Ag/AgCl, a nickel-rich deposit featuring a Ni/Co ratio of 1.81 was formed with pristine copper in concentrated chloride, and it increased to 7.05 by employing PDADMA/Cu with the polymer loading of 0.75 mg cm$^{-2}$ (Fig. 4a). Similarly, at a cobalt-favorable potential of −0.725 V vs Ag/AgCl, concentrated chloride showed already superior cobalt selectivity without PDADMA (Co/Ni: 14.08 at 100 mM Co(II) + Ni(II)) owing to the anomalous electrodeposition, and the thin PDADMA layer (0.07 mg cm$^{-2}$) brought about further enhancement, reaching the highest surface Co/Ni ratio of 16.73 on electrodeposit (Fig. 4b). For both cobalt (−0.725 V vs

Ag/AgCl) and nickel (−0.6 V vs Ag/AgCl), polymer-driven selectivity enhancement was observed only in concentrated chloride thanks to the speciation control, while other electrolytes with low-to-moderate chloride did not display significant PDADMA-driven improvements (Fig. 4a, b). In low-to-moderate chloride concentrations, both Co(II) and Ni(II) mainly exist as a cationic complex, and therefore the positive polyelectrolyte PDADMA has no charge-specific stabilization effect at the interface toward the metals. Thus, these results prove the need for both speciation control through electrolyte selection and interface tuning by surface functionalization to achieve the desired synergistic effect.

Solid-phase surface characterizations based on spectroscopy further supported the results in Fig. 4a, b: the high surface Co/Ni ratio was confirmed by X-ray fluorescence (XRF) analysis (Co/Ni: 16.0, Fig. 4c) and energy-dispersive spectroscopy (EDS) mapping (Co/Ni: 18.4, Fig. 4d–f), supporting the observation that PDADMA/Cu has larger Co/Ni ratios compared to pristine Cu (Supplementary Figs. 11 and 12). X-ray photoelectron spectroscopy (XPS) analysis also confirmed PDADMA-driven selectivity improvements for both nickel-rich deposits at −0.6 V vs Ag/AgCl (Ni/Co: 1.45 and 3.01 for pristine Cu and PDADMA/Cu (0.75 mg cm$^{-2}$), respectively) and cobalt-rich deposits at −0.725 V vs Ag/AgCl (Co/Ni: 10.19 and 12.41 for pristine Cu and PDADMA/Cu (0.07 mg cm$^{-2}$), respectively) (Supplementary Fig. 13). In addition, the peak fitting of XPS analysis exhibited several peaks at the binding energy of: ~778.3 eV (metallic Co), ~780.5 eV (CoOOH), and ~782.0 eV (Co(OH)$_2$) in the Co 2$p$3/2 spectrum and ~852.5 eV (metallic Ni), ~855.6 eV (Ni(OH)$_2$), and ~856.5 eV (Ni$_2$O$_3$) in the Ni 2$p$3/2 spectrum (Supplementary Fig. 13)[54–57]. These peaks indicated the formation of oxide/hydroxides, including metallic species, probably due to surface oxidation and/or hydroxide formation during electrodeposition, which in part agreed with EQCM analysis (Eqs. (2) and (3)).

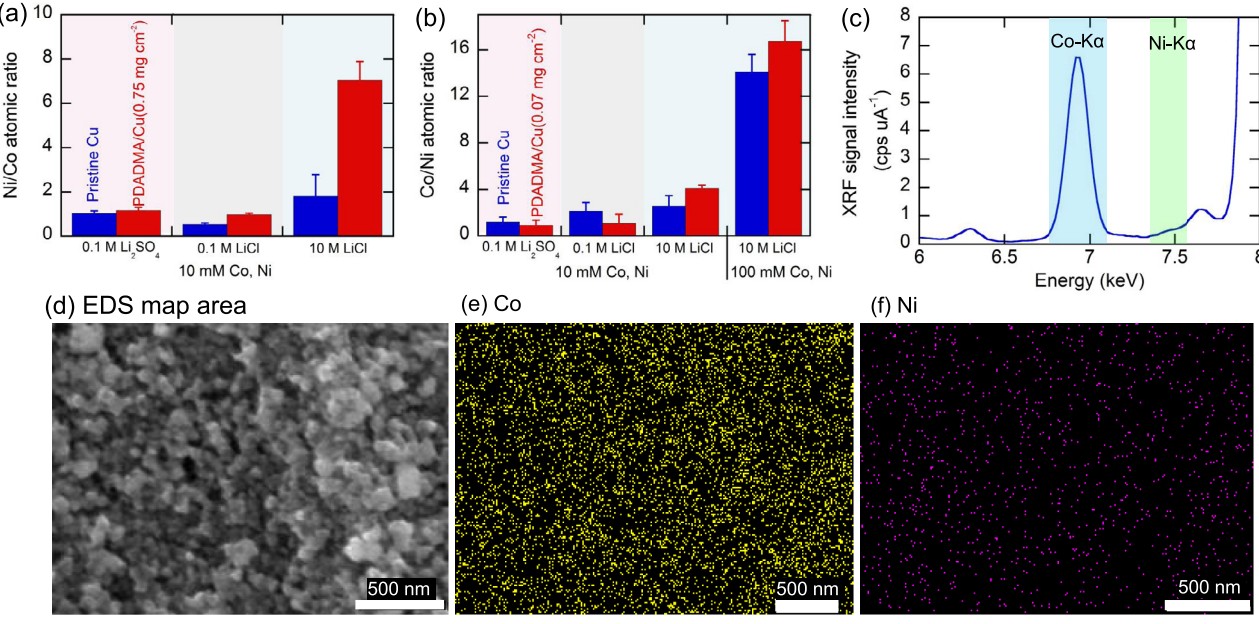

**Fig. 4 Synergistic effect of electrolyte and interfacial control.** The effect of a background electrolyte and interfacial polymer on **a** surface Ni/Co ratio at nickel-favored potential of −0.6 V vs Ag/AgCl and **b** surface Co/Ni ratio at cobalt-favored potential of −0.725 V vs Ag/AgCl. **c** A XRF spectrum and **d–f** EDS mapping of cobalt and nickel on the electrodeposit formed using PDADMA/Cu (PDADMA loading: 0.07 mg cm$^{-2}$) at −0.725 V vs Ag/AgCl in 100 mM Co(II) and Ni(II) in 10 M LiCl. Scale bars for EDS and cobalt/nickel mapping are 500 nm. Error bars indicate standard error of the mean ($n = 3$).

Generating oxides/hydroxides as final products readily provides value-added, recyclable precursors for the fabrication of cathode materials[58–60]. In addition, the morphology of anomalously electrodeposited cobalt and nickel in the presence of PDADMA differed from the deposit formed in the absence of PDADMA. As shown in Supplementary Fig. 14a, the deposit without PDADMA exhibited the formation of needle-like dendrites. The formation of dendrites can be ascribed to a locally enhanced electric field, as observed in the literature[61]. In contrast, PDADMA/Cu (0.07 g cm$^{-2}$) exhibited rough and grainy deposits (Supplementary Fig. 14b), and thicker PDADMA/Cu (0.75 mg cm$^{-2}$) showed wrinkled morphology (Supplementary Fig. 14c), without sharp dendrites. The scanning electron microscopy (SEM) analysis revealed that PDADMA tuned not only cobalt to nickel selectivity but also affected the morphology of electrodeposited metal, which can be ascribed to the surface conduction of CoCl$_4^{2-}$ in the positively charged PDADMA layer[61–63]. Future studies will pursue the elucidation of the detailed mechanisms for dendrite control on PDADMA-coated electrodes in cobalt and nickel electrodeposition, with potential applications in materials processing.

Also, the reversible nature of electrodeposition and stripping of cobalt and nickel was demonstrated by first electrodepositing in 100 mM Co(II) + Ni(II) in 10 M LiCl at -0.725 V vs Ag/AgCl and then by applying −0.08 V vs Ag/AgCl for releasing (stripping) electrodeposited cobalt and nickel into 5 mM NaNO$_3$, whose pH was adjusted to 2.9–3.0 (Supplementary Figs. 15 and 16). A current–time ($I$ vs $t$) plot during stripping revealed a gradual decrease in the magnitude of the electrochemical current, which can be associated with the rapid dissolution of electrodeposited cobalt/nickel (Supplementary Figs. 15b and 16b). As depicted in Supplementary Figs. 15c and 16c, PDADMA/Cu electrodes exhibited high stripping efficiencies (>90%, defined as the ratio of cobalt/nickel stripped to electrodeposited) for both metals, with the same trend of selectivity tuning observed. We also employed EQCM analysis, allowing the direct tracking of the change in potential, current, and mass on the quartz crystal in real time during the electrodeposition/stripping (Supplementary

Figs. 17–19). First, when using a Cu-coated quartz crystal (Supplementary Fig. 17), applying −0.725 V vs Ag/AgCl revealed the increase in the mass (70 ng s$^{-1}$), which was ascribed to the cobalt/nickel electrodeposition. We observed that the deposited cobalt/nickel was released into 5 mM NaNO$_3$ (pH = 2.9–3.0) by applying −0.08 V vs Ag/AgCl; about 96% of the mass increase caused by electrodeposition was recovered during the stripping phase (Supplementary Fig. 17c), indicating high reversibility of electrodeposition/stripping. Also, we tested the similar EQCM analysis during electrodeposition/stripping using PDADMA/Cu- and PDADMA/Au-coated quartz crystal to confirm the stability of PDADMA (Supplementary Figs. 18 and 19). We observed an initial drop in the mass of the electrode during open-circuit pre-equilibrium stage (Supplementary Figs. 18c and 19c), which was attributed to the dissolution of PDADMA in the deposition electrolyte; however, the dissolution of the polymer did not continue and there appeared a plateau in the mass. During the electrodeposition, there was a discernable increase in the mass of the electrode (average rate: 6.8 ng s$^{-1}$ for PDADMA/Cu and 5.9 ng s$^{-1}$ for PDADMA/Au); here the obtained mass exceeded the theoretical maximum mass increase calculated with the assumption of 100% faradaic efficiency, indicating the re-adsorption of the positively charged PDADMA onto the cathodic substrate. Also, the chronoamperometric stripping revealed a current peak in the current–time curve (Supplementary Figs. 18b and 19b), which can be associated with the simultaneous stripping of cobalt/nickel, as reflected in the decrease in the mass during the corresponding time interval of the peak (Supplementary Figs. 18c and 19c). The change in the mass approached to zero immediately after the anodic current diminished to zero. There was a slight, continuous decrease in the mass on the quartz crystal (average rate: −1.33 ng s$^{-1}$ for PDADMA/Cu and −0.72 ng s$^{-1}$ for PDADMA/Au) even after stripping was over (that is, after the current became stabilized), due to the electrostatic repulsion between the substrate and positively charged PDADMA. Even so, this mass loss can be effectively prevented by stopping the chronoamperometric operation once

the stripping current approaches zero. In our experiments, the mass loss of PDADMA upon prolonged anodic stripping (1 h) accounted for only <0.3% of the entire PDADMA loading (0.75 mg cm$^{-2}$), demonstrating the stability of the polyelectrolyte upon electrodeposition/stripping under the given electrolyte conditions. Thus, our results point to an innovative way of recovering electrodeposited cobalt/nickel without the intensive use of harmful chemicals.

**Recovery of cobalt and nickel from spent NMC cathodes.** Our findings provide fundamental insights on how synergistic electrolyte and surface charge control can tune selectivity during electrodeposition of two metals with similar reduction potentials. Beyond fundamental studies in interfacial electrochemistry, we envision this concept to be applicable to the selective recovery of cobalt and nickel from spent LIBs, providing a sustainable pathway for battery recycling. To provide a proof of feasibility, we pretreated commercial 18650 3 Ah graphite‖NMC battery cells through discharging, dismantling, N-methylpyrrolidine (NMP) treatment, and leaching (Fig. 5a, see the "Methods" section for details) to separate cathode active materials from the other electrode components—more precisely, from aluminum current collector and polyvinylidene fluoride (PVDF) binder—in a safe and efficient manner. We leached 4 g of harvested cathode powder (obtained after NMP treatment/filtration/drying) in 30 mL of 10 M HCl, and pH was adjusted to 3.0 using LiOH; this procedure resulted in the formation of a dark green mixture of nickel-rich concentrated chloride, composed of cobalt (5,695 mg L$^{-1}$), nickel (37,150 mg L$^{-1}$), and manganese (2,820 mg L$^{-1}$)—the molar ratio of Co:Ni:Mn was 1.00:6.52:0.50.

Here we experimentally demonstrated the feasibility of the designed electrochemical recovery process for battery recycling by tackling leached NMC cathodes as the feedstock (Fig. 5b). First, a cycle of electrodeposition/stripping was carried out: first electrodeposition at −0.725 V vs Ag/AgCl on a PDADMA/Cu (0.07 mg cm$^{-2}$) electrode allowed for selective up-concentration of cobalt on an electrodeposit (stream A in Fig. 5b), and anodic stripping provided a simple way of releasing recovered solid-phase cobalt/ nickel into a liquid phase for secondary up-concentration and processing. Inductively coupled plasma optical emission spectroscopy (ICP-OES) analysis revealed that the molar ratio of Co:Ni:Mn in the stripping electrolyte changed to 1.00:0.60:0.02 (Fig. 5c). Also, the stripping electrolyte, after the addition of 10 M LiCl, exhibited a distinctively bluish color, which originates from the formation of predominant CoCl$_4^{2-}$ complex, as compared to the strong greenish and nickel-rich electrolyte obtained right after the leaching (Fig. 5d), again confirming the up-concentration of cobalt over nickel by electrodeposition/stripping. The secondary PDADMA-driven electrodeposition at −0.725 V vs Ag/AgCl from the up-concentrated electrolyte brought about significantly improved cobalt purity (stream C), reaching 96.4 ± 3.1% (Fig. 5c).

Also, the first selective cobalt deposition led to an increase in Ni/Co ratio in the remaining liquid phase (stream B in Fig. 5b). The molar ratio of Co:Ni:Mn in the liquor B after the first cobalt electrodeposition was 1.00:9.05:0.73, leading to an advantageous condition for subsequent selective nickel recovery. At −0.6 V vs Ag/AgCl, nickel purity of 94.1 ± 2.3% on the deposit (stream D) was obtained after 30 min electrodeposition (Fig. 5c). The remaining liquid electrolyte after the secondary cobalt electrodeposition (stream E) can be fed into a cobalt-selective

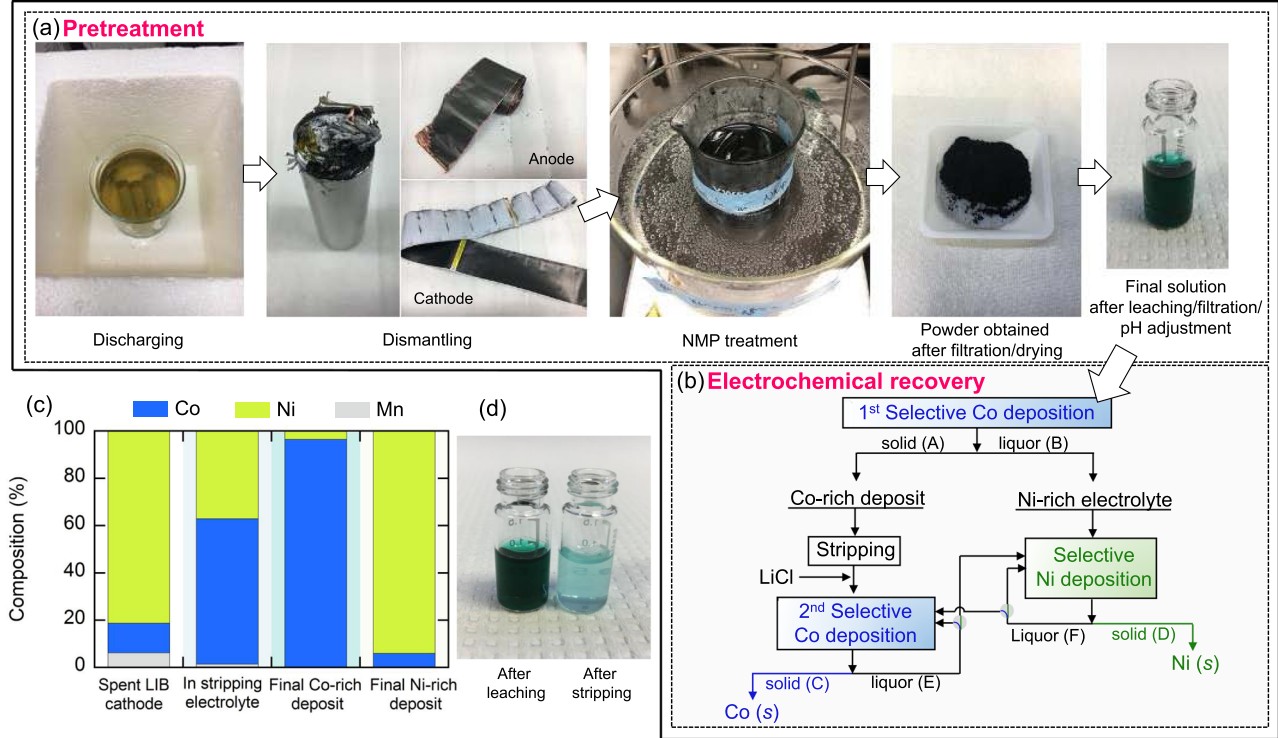

**Fig. 5 Application of selective electrodeposition for potential use in battery recycling processes. a** Pretreatment steps of spent LIBs: discharging, dismantling, NMP treatment, and leaching. See the "Methods" section for details. **b** A simplified schematic representation of the process envisioned in this work for the electrochemical recovery of cobalt and nickel. **c** Molar composition of liquid/solid hydrometallurgical streams: (i) spent NMC cathode, (ii) after selective cobalt electrodeposition followed by anodic stripping, (iii) final Co-rich electrodeposit after second cobalt-selective electrodeposition, and (iv) final Ni-rich electrodeposit after selective nickel electrodeposition. **d** Pictures of the liquid electrolytes: (i) after leaching and (ii) after a cycle of selective cobalt deposition/stripping and the addition of 10 M LiCl.

electrodeposition unit for multiple cycles until desired cobalt purity/recovery rate are obtained. If the Co/Ni ratio in the stream E becomes reversed (e.g., Co/Ni < 1), which is not a desirable condition for the recovery of high-purity cobalt, the stream can be sent for selective nickel recovery to control nickel concentration. In a similar way, selective nickel deposition can be conducted for multiple cycles from the stream F, and as the level of nickel decreases and thus the ratio Co/Ni becomes too high, which is disadvantageous for high-purity nickel recovery, the stream can be diverted for selective cobalt deposition. The two selective deposition processes can work in a complementary manner to control Co/Ni ratio to be in an appropriate level. Also, during the selective recovery of cobalt/nickel assisted by PDADMA, there was negligible manganese co-deposition due to the large potential window (Supplementary Fig. 20).

Based on the recovery result using the cylindrical Li-ion cells, we carried out a technoeconomic analysis (TEA) by considering the market prices of the reagents, products, and energy consumptions involved in this process (see the Supplementary Information for the TEA framework utilized). Even with the assumption of 0.3% polymer loss per a cycle of deposition/stripping (based on EQCM analysis, Supplementary Figs. 18 and 19), the cost of PDADMA in the entire process turned out to be only 3.61% of the cost benefit from harvesting cobalt and nickel, and consisted in only 0.41% of the entire material costs (Supplementary Table 2). The major material cost is derived from the usage of lithium hydroxide, which is required for pH adjustment after HCl leaching. This challenge could be resolved by recovering and recycling this expensive lithium salt in the form of LiCl, as proposed in our calculations (Supplementary Fig. 21). Consequently, a revenue in materials flow was estimated to be about $2.230 kg$^{-1}$ by assuming 95% metal recovery from a kilogram of NMC powder (Supplementary Table 2). The energy consumption analysis was subsequently conducted with 95% metal recovery scenario (Supplementary Table 3). The main energy consumption in this proposed process comes from the final harvesting of LiCl, especially in the final electrodeposition and drying stage, but the drying cost is expected to be significantly decreased given that heat energy can be more efficiently utilized at a larger scale. The energy consumption and the cost of the electrical energy in the whole process were estimated to be 29.4 kWh kg$^{-1}$ and $2.027 kg$^{-1}$, respectively, based on our unoptimized laboratory-scale results (Supplementary Table 3). Finally, the final profit in the whole process, with material revenue, material cost, and energy consumption all considered, was found to be $0.2 kg$^{-1}$ waste NMC powder based on 95% metal recovery, which is comparable to solvent extraction/precipitation-based NMC recycling techniques[64], with the possibility of further optimization in electrolyte recycling strategy (e.g., drying). While this analysis proves that valorization of waste metals via selective electrodeposition could be a viable option in the future, there are major hurdles that need to be overcome before larger-scale implementation. For example, a significant improvement in the selectivity and recovery rate in the unit operation is required to be able to be competitive as compared to other state-of-the-art techniques (Supplementary Table 1). Further in-depth systematic parametrization studies are currently underway to improve selectivity and recovery rate by uniform coating of PDADMA polymer and rational design of electrochemical interfaces and electrochemical cells.

In summary, we have successfully shown that speciation control through electrolyte engineering, combined with surface functionalization by using charged polymers, enabled the synergistic tuning of metal selectivity during electrochemical deposition. The use of concentrated chloride imparted opposite charges to cobalt and nickel, and thus discriminated between

metals with otherwise similar reduction potentials and ionic characteristics, allowing for potential-dependent selectivity by leveraging anomalous deposition—with the electrolyte being able to be recycled. When even small amounts of positive polyelectrolyte PDADMA were coated on the substrate, enhanced selectivity was achieved due to the controlled mobility of $CoCl_4^{2-}$ in the layer of the polymer matrix. We applied our proposed process for the direct recovery of cobalt and nickel from practical spent NMC cathodes, demonstrating that high-purity metal recovery can be achieved solely by electrochemical pathways. We envision that the recovered cobalt and nickel can be possibly revalorized as value-added precursors for the fabrication of fresh cathode materials. In the future, we expect our findings on electrodeposition at functional surfaces to not only enable selectivity for precision metal separations, but also offer pathways for materials processing through morphology control and patterning.

## Methods

**Electrodeposition of cobalt and nickel**. To highlight the synergistic contributions of speciation control and interfacial tailoring on selectivity, we initially explored solely the effect of electrolyte by using a copper foil as a cheap and conductive substrate without the use of the polymer coating. Next, we investigated how the combined use of controlled electrolyte and polymer interfaces overcomes the limitations of conventional electrodeposition, and offers selectivity tunability.

All the electrochemical deposition tests were conducted in a BASi (Bioanalytical Systems, Inc.) electrochemical cell (Supplementary Fig. 22) with a three-electrode configuration. Copper foil was employed as a working electrode for cobalt and nickel deposition; the electrodes were prepared by cutting the copper foil (thickness 0.25 mm, 99.98% trace metals basis, Sigma-Aldrich) into a dimension of 1 cm × 2 cm. Copper foil was thoroughly washed with ethanol (>99.9% purity, <0.1% water content) and acetone (>99.5% purity, <0.5% water content) before use. Then the back side of the foil was pasted on electrical tape (3 M Scotch$^{TM}$ Super 33+ Vinyl Electrical Tape, thickness: 0.177 mm). For the preparation of PDADMA-coated copper foil, 0.75 µL of PDADMA solutions with different concentrations (0.1, 0.5, 1, 5, 10, 20 mg PDADMA in 1 mL of ethanol/de-ionized water (1/1, v/v)) were drop-casted on pristine copper substrates and dried for 10–12 h in ambient air and at room temperature (21–23 °C). The effective working area of cobalt and nickel deposition, immersed in the electrolyte, was 0.5 cm$^2$. A platinum wire (length: 7.5 cm, diameter: 0.5 mm, purity: 99.95%), which was isolated from the bulk electrolyte by a glass body and porous CoralPor$^{TM}$ tip, was used as a counter electrode. A reference electrode of Ag/AgCl in 3 M NaCl was used. Electrochemical tests were carried out using LSV and chronoamperometry using a potentiostat/galvanostat (VSP-300, BioLogic) in ambient air and at room temperature (21–23 °C), without the use of argon- or nitrogen-filled glove box. For the electrodeposition, 3 mL of the electrolyte, which contained $CoCl_2$ ($CoSO_4$) and/or $NiCl_2$ ($NiSO_4$) as metal sources in different background electrolytes (0.1 M $Li_2SO_4$, 0.1 M LiCl, and 10 M LiCl) were purged with nitrogen (purity: >99.99%) before the test. When using simulated solutions, the initial concentration of the binary cobalt/nickel was 10 or 100 mM; ensuring selectivity in diluted conditions. Despite mass-transfer limitations and thus suffering from concentration polarization, this technique demonstrates a broad applicability for various solid/liquid ratios in the leaching process. In the LSV test, the onset potential of electrodeposition was defined as the intersection of tangential lines of the horizontal background current (non-faradaic zone) and the faradaic zone in the initial current increase.

**Quantification of electrodeposited cobalt and nickel**. To recover the metals for elemental analysis, the electrodeposits were thoroughly washed with de-ionized water and then digested using 10% w/w $HNO_3$. The amount of electrodeposited cobalt and nickel was quantified using ICP-OES (Agilent 5110). In all cases, 2% w/w $HNO_3$ was used to dilute samples of calibration standards or solutions generated after electrodeposition and digestion. Standard solutions of 100, 500, 1000, and 5000 ppb cobalt/nickel were prepared by diluting the ICP calibration standards (cobalt/nickel standard for ICP TraceCERT®, 1000 mg/L in nitric acid, Sigma-Aldrich) with 2% w/w $HNO_3$ (with 2% w/w $HNO_3$ being blank). After calibration, the linear fit was visualized, ensuring $R^2$ of >0.999 for every measurement. Each sample was measured with at least 15 replicates by the spectrometer to yield a reliable averaged reading. From the ICP measurements, faradaic efficiencies of metal electrodeposition were determined by:

$$\text{Faradaic efficiency} = \frac{n \times M \times F}{Q_{total}} \times 100$$

where $M$ (mol) is the sum of the amount of electrodeposited cobalt/nickel determined by ICP-OES, $F$ is the Faraday constant (96,485 C mol$^{-1}$), $Q_{total}$ is the total charge passed through during the electrodeposition, and $n$ is the number of

electrons involved in cobalt/nickel electrodeposition. Considering that two electrons are involved either in direct deposition (Eq. (1)) or through hydroxide formation (Eqs. (2) and (3)), $n = 2$ was used for the determination of faradaic efficiency.

**Stripping of electrodeposited cobalt and nickel**. After electrodeposition, the electrode with the electrodeposit was transferred to a stripping electrolyte of 5 mM NaNO$_3$, whose pH was adjusted to 2.9–3.0 using 12 M HCl. In this weak acid, a pristine copper foil exhibited an equilibrium potential of −0.04 to +0.02 V vs Ag/AgCl. Thus, applying a potential of −0.08 V vs Ag/AgCl did not lead to anodic copper dissolution, but allowed the anodic stripping of the electrodeposited cobalt and nickel. Stripping was continued until the anodic current became <10 μA. The amount of the recovered cobalt and nickel in the stripping electrolyte was measured using ICP-OES analysis. Also, the amount of remaining cobalt and nickel on the electrodeposit after the stripping was determined by digesting the deposit and quantifying the metals using ICP-OES, as described above. Finally, stripping efficiencies were determined by:

$$\text{Stripping efficiency} = \frac{m_{\text{stripped}}}{m_{\text{stripped}} + m_{\text{deposited}}} \times 100,$$

where $m_{\text{stripped}}$ (mol) is the amount of stripped cobalt or nickel and $m_{\text{deposited}}$ (mol) is the amount of remaining cobalt or nickel in the electrodeposit.

**EQCM analysis**. In situ electrochemical gravimetric analysis was carried out using a working electrode of 5 MHz quartz crystal coated with Cu, with a piezo-electroactive area of 0.2 cm$^2$ (diameter: 14 mm, polished finish, AW-R5CUP, BioLogic). The counter electrode was a platinum wire, and all the potentials are referenced to Ag/AgCl (in 3 M NaCl) electrode. The frequency shift was measured using EQCM (BioLogic BluQCM QSD (QSD-TCU)). The mass increase was determined using the Sauerbrey equation[27]:

$$\Delta f = \frac{-2f_0^2 \Delta m}{A\sqrt{\mu_i \rho_i}} = -K\Delta m$$

where $f_0$ is the resonant frequency of the quartz crystal, $A$ is the piezoelectroactive area, $\mu_i$ is the shear modulus of the quartz ($2.947 \times 10^{11}$ g cm$^{-1}$ s$^{-2}$) and $\rho_i$ is density of the quartz (2.648 g cm$^{-3}$).

**Characterization**. Materials characterization was conducted in the Frederick Seitz Materials Research Laboratory Central Research Facilities, University of Illinois. The surface morphology imaging and elemental mapping images after electrodeposition were obtained using a SEM (Hitachi S-4700) operated at an accelerating voltage of 5 kV, equipped with EDS (iXRF) with an accelerating voltage of 15 kV. The chemical states of cobalt and nickel on the electrodes were characterized using XPS (Kratos Axis ULTRA) with monochromatic Al Kα X-ray source (210 W). The XPS results were analyzed using the CASA XPS software (UIUC license). XRF (Shimazdu EDX-7000 energy-dispersive XRF spectrometer) was run under helium atmosphere, using a rhodium target with accelerating potential up to 50 kV; integration times were 100 and 500 s for qualitative–quantitative and quantitative scans, respectively. Ultralene film was used to support the samples, and collimator sizes were 3–10 mm. PCEDX-Navi software was used for data processing and analysis.

**Pretreatment and leaching of end-of-life spent LIBs**. Unused 18650 batteries (Hohm Tech Life V4 18650 3015 mAh 22.1 A) were obtained from Hohm Tech. The following pretreatment steps were conducted before the electrochemical recovery was carried out:

(a) Discharging: The batteries were immersed in 10% (w/v) NaCl for 24 h to completely discharge. The remaining cell voltage was frequently monitored using a portable multimeter and full discharge was confirmed before manual disassembly.
(b) Dismantling: The batteries were manually disassembled using a saw and a sharp-nosed plier in an as-installed fume hood, and anode/cathode materials were uncurled for separation. The cathode scraps were cut into small pieces (1 cm × 1 cm).
(c) NMP treatment: The cathode active materials were separated from aluminum current collector by employing NMP as a solvent to dissolve PVDF binder. The small pieces of cathode scraps were treated in NMP at 100 °C for 24 h. Afterwards, the cathode materials were vacuum-filtered using a Buchner funnel and a filter paper (Whatman® grade 541) and dried at 140 °C using a drying and heating chamber (Binder, Model FD 115) with forced convection.
(d) Leaching: All the leaching experiment were conducted in a 250 mL Erlenmeyer flask at room temperature. In all, 30 mL of 10 M HCl was poured into the reactor. Four grams of the filtered cathode materials were then slowly added to the reactor and stirred continuously at 300 rpm for 2 h. After leaching, the insoluble residue was separated by filtration, and the concentrations of Co, Ni, and Mn were determined using ICP-OES.

## Data availability
All experimental data reported in this study and Supplementary Information are available from the corresponding author upon reasonable request.

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

## Acknowledgements

The information, data, or work presented herein was funded in part by the Advanced Research Projects Agency-Energy (ARPA-E), U.S. Department of Energy, under Award Number DE-AR0001396. The fundamental studies within the work was also partly supported by U.S. Department of Energy, Office of Basic Energy Sciences under Award Number DOE DE-SC0021409.

## Author contributions

K.K. and X.S. conceived and designed the experiments, K.K. performed the experiments and analyzed the data. K.K., D.R., and R.C. contributed materials/analysis tools. X.S. supervised the project. K.K. wrote the paper, and all authors contributed to the editing and review of the manuscript.

## Competing interests

The authors declare no competing interests.
