## [Peer Review File · Nature Communications]

Reviewers' comments:

Reviewer #1 (Remarks to the Author):

Su et al. report on an electrochemical method to separate cobalt, manganese and nickel ions from the same solution. This is of high importance for recycling of Li ion batteries where cobalt is still the most important element and the NMC-based LIBs (having nickel, manganese and cobalt) still dominate. From this perspective, current recycling approaches covering industrial and academic efforts cannot be considered as sustainable and do involve the formation of toxic substances (in situ) during the recycling process and are connected to high energy consumption. In addition, the purity of recovered materials is lower than the pristine materials.

In their current manuscript, Su have addressed the separation of Co from simulated solutions, which has the highest price and it is therefore the most important element to be recycled from the economic perspective, by using electrochemical deposition. Instead of only focusing on electrolyte and metal complex engineering, the authors have also modified the electrode where electrodeposition takes place. This combined approach allows to recover Co with a purity of app. 96%. Although the purity is worse than for other selected methods for Co recovery (adsorption on ion-selective sorbents, e.g. MOFs or similar), the here presented method can be more easily scaled up which renders it very promising and provides significant novelty. However, the manuscript does not relate this Co recovery efficiency to other literature reports, making it thus impossible to compare these results if only reading the manuscript. The authors only briefly state: "Solvent extraction, ion-exchange, precipitation, and membrane-based separations have been proposed for the separation of cobalt and nickel¹⁰, but they suffer from large chemical input with a long refining processes and complicated solution/speciation chemistry." In fact, there has been recently significant progress on recycling of Li ion batteries and in particular green methods for cobalt recovery. A recent review summarizes for example the state of the art in cathodes (especially Co) recovery and outlines electrochemical metal recovery methods: (Adv. Energ. Mater. <https://doi.org/10.1002/aenm.202003456>). The authors should revise the introduction accordingly and relate their results to the current state of the art in the field.

The methodology (involving ICP and XPS) and data analysis of the manuscript is detailed and support sufficiently the claims, but I could not find SEM images of the electrode before and after polymer coating in order to judge whether the polymer coating is uniform. The authors should add this comparison based on SEM and check whether there are cracks in the polymer coating on the surface. Could this be a reason for not achieving complete Co (near 100%) separation?

This is a manuscript with important results for Co recovery because it presents an energy-efficient and scalable method, but the weak point is the lack of comparison to literature reports to give an overview on the current state of the art in the field. This requires that the manuscript undergoes a major revision before it could be accepted in Nat. Comm. I would be happy to review the revised version of the manuscript.

Reviewer #2 (Remarks to the Author):

This paper intends to achieve selectivity for cobalt and nickel with the synergistic combination of electrolyte control and interfacial design in the potential-dependent electrodeposition. The chemical properties of nickel and cobalt are similar, prompting the industry to use cheap nickel to replace part of the cobalt in lithium-ion batteries (LIBs), which makes it difficult to achieve a high-level recycling of nickel and cobalt waste in spent LIBs, simultaneously. This article focuses on the separation of nickel and cobalt, which is very interesting, but there are still some problems in the research itself. I will

consider his official publication only after a major revise.

1. The concentration of Co and Ni in the simulated solution is only 10 mM, it seems this concentration is much lower than the real situation in presently recovery process. It would be better to add more explanation about the representativeness of the solution.
2. According to the Fig 2(f), it indicated that a higher Co/Ni ratio could be achieved when the potential is -0.75 V, but in the following text and experiment, the potential of 0.725 was used, what's the reason for this difference? Meanwhile, in the Fig 2 e and f, the results should be present as line chart rather than a smooth curve.
3. In Fig. 3A, when the potential is -0.725V, if the PDADMA loading is higher than 5 mg/cm², can selectivity be further improved?
4. The results of the first half of the paper are the experiments carried when Co and Ni are in the same concentration. But when the electro deposition is carried out following the proposed chart, will the different concentrations of the two elements have a significant impact on the recycling performance?
5. In Fig. 5, Why not electrodeposition at 0.6V first and then electrodeposition at 0.725? What is the voltage control sensitivity that can be realized in industrial practice? Is it operational?
6. Whether the technology can be applied in practice depends on its environmental hazards and economic value. The author must supplement the LCA-based environmental impact assessment and economic assessment of this technology, so as to prove that the technology has published value.

Reviewer #3 (Remarks to the Author):

This paper proposes the use of electrodeposition to separate nickel from cobalt in waste streams produced from lithium ion batteries. In principle some of the science is passable although I was not surprised by the results. There is no clear urgency to publish these results. The authors unfortunately display a lack of knowledge about lithium ion battery recycling. From the perspective of kinetics, cost, digestion and efficiency this methodology is totally impractical. The work could be publishable in a much lower impact journal if discussion of lithium ion battery recycling was removed. This work should not be published in any Nature journal.

We thank all the reviewers for the thoughtful comments, and valuable suggestions to improve our manuscript. We have revised our manuscript to include extensive new experiments, to (i) demonstrate the feasibility of our system for practical LiB recovery using real LiB cathodes (including disassembling and recovering of realistic Co/Ni concentrations from the full leaching and pre-treatment steps), (ii) strengthen our introduction with state-of-art work across different fields in critical element recovery, and finally, (iii) provide a comparative technoeconomic analysis of our system to help insights into advantages and possible limitations. We have also added additional SEM and supporting experiments to address the reviewer suggestions. The point-by-point response to the reviewer suggestions are given below in blue.

Reviewers' comments:

Reviewer #1 (Remarks to the Author):

Su et al. report on an electrochemical method to separate cobalt, manganese and nickel ions from the same solution. This is of high importance for recycling of Li ion batteries where cobalt is still the most important element and the NMC-based LIBs (having nickel, manganese and cobalt) still dominate. From this perspective, current recycling approaches covering industrial and academic efforts cannot be considered as sustainable and do involve the formation of toxic substances (in situ) during the recycling process and are connected to high energy consumption. In addition, the purity of recovered materials is lower than the pristine materials.

In their current manuscript, Su have addressed the separation of Co from simulated solutions, which has the highest price and it is therefore the most important element to be recycled from the economic perspective, by using electrochemical deposition. Instead of only focusing on electrolyte and metal complex engineering, the authors have also modified the electrode where electrodeposition takes place. This combined approach allows to recover Co with a purity of app. 96%. Although the purity is worse than for other selected methods for Co recovery (adsorption on ion-selective sorbents, e.g. MOFs or similar), the here presented method can be more easily scaled up which renders it very promising and provides significant novelty. However, the manuscript does not relate this Co recovery efficiency to other literature reports, making it thus impossible to compare these results if only reading the manuscript. The authors only briefly state: "Solvent extraction, ion-exchange, precipitation, and membrane-based separations have been proposed for the separation of cobalt and nickel, but they suffer from large chemical input with a long refining processes and complicated solution/speciation chemistry." In fact, there has been recently significant progress on recycling of Li ion batteries and in particular green methods for cobalt recovery. A recent review summarizes for example the state of the art in cathodes (especially Co) recovery and outlines electrochemical metal recovery methods: (Adv. Energ. Mater. <https://doi.org/10.1002/aenm.202003456>). The authors should revise the introduction accordingly and relate their results to the current state of the art in the field.

We thank the reviewer #1 for insightful comments and suggestions on comparative technologies, which definitely strengthen the quality of our manuscript. Following the reviewer's suggestions, we modified the *introduction* part accordingly to: (i) explain the battery recycling process (pretreatment, discharging, dismantling, separating, leaching and recovery), (ii) emphasize the necessity of cobalt/nickel separation from post-leaching solution, (iii) give an overview of commercialized and state-of-the-art technologies based on various methods, including solvent

extraction, precipitation, and adsorption. Also, we summarized the state-of-the-art green technologies for hydrometallurgical cobalt/nickel separations that can be beneficial for metal revalorization from NMC chemistry, with special focus on selectivity performance metrics (**Table S1**).

The following paragraph was added in the introduction.

“Considering the high content of valuable d-block elements, the recycling of multi-metallic cathodes, such as lithium nickel manganese cobalt oxide (NMC) cathode, has been paid particular attention. In general, hydrometallurgical processes for cathode recycling involve a series of pretreatment steps, including discharging, dismantling, separating, and harvesting of active materials from an current collector¹⁰. In a subsequent leaching step, the constituent elements in a solid-phase are transferred into a liquid-phase for further purification. The selective separation of cobalt and nickel from post-leaching solution is imperative to ensure a sustainable method of recovering each constituent metal with high purity, but it is challenging due to the similar physicochemical properties between cobalt and nickel. State-of-art recycling processes (e.g., LithoRec process, a lab-scale process by Aalto University) rely on solvent extraction, precipitation, or a combination of these as a separation mean of cobalt and nickel¹¹. Also, there have been extensive studies at a lab-scale for the separation of cobalt and nickel, such as solvent extraction^{12, 13}, precipitation¹⁴, adsorption^{15, 16, 17, 18, 19, 20}, intercalation²¹, and dialysis²², all of which can be beneficial for cobalt/nickel recovery in NMC chemistry regime. A comparison of different state-of-the-art cobalt/nickel separation techniques are summarized in Table S1, to provide benchmarks for selectivity. Of particular note, solvent extraction and precipitation usually exhibit high selectivity performance, but often can incur large chemical costs or waste, and may face challenges concerning complex solution/speciation chemistry^{9, 23}. As such, new technologies that can complement or assist in process intensification of these complex purification trains are urgently needed, especially if they can lower either thermal/chemical consumption or waste generation.”

Table S1. Comparison of various state-of-the-art techniques for separation of cobalt and nickel based on selectivity performance metrics.

Technique	Co, Ni concentration	Leaching solution/background electrolyte	Key materials (precipitant/extractant /adsorbent/electrode)	Selectivity performance metric	Refs
Precipitation	[Co]: 9.05 g L ⁻¹ [Ni]: 4.34 g L ⁻¹ (after Mn recovery)	3 M H ₂ SO ₄ + 3 vol% H ₂ O ₂	Ni: C ₄ H ₈ N ₂ O ₂ Co: NaOH	100 Ni over Co separation factor ^a (~48 Ni over Co on precipitate)	1
Solvent extraction	[Co]: 15 g L ⁻¹ [Ni]: 21 g L ⁻¹	4-8 M HCl	[P ₈₈₈₈][oleate]	30,000 Co over Ni separation factor ^b	2
Solvent extraction	[Co]: 14 g L ⁻¹ [Ni]: 15 g L ⁻¹	2 M H ₂ SO ₄ + 6 vol% H ₂ O ₂	Cyanex 272	750 Co over Ni separation factor ^b	3
Adsorption	[Co]: 2.10 ppm [Ni]: 1.98 ppm	4 M H ₂ SO ₄ + 30wt% H ₂ O ₂	(E)-4-[(2- mercaptophenyl)diazenyl] -2-nitrosophthalen-1-ol in γ-Al ₂ O ₃ monoliths	62.7 Co over Ni separation factor ^c	4
Adsorption	[Co]: 2.10 ppm [Ni]: 10.1 ppm	4 M H ₂ SO ₄ + 30wt% H ₂ O ₂	[(E)-4-((3-amino-4- hydroxyphenyl)diazenyl)n aphthalen-1-ol (AHPDN)] in platelets of ZnO	17.1 Co over Ni separation factor ^c	5

Adsorption	[Co]: 5 $\mu\text{g mL}^{-1}$ [Ni]: 5 $\mu\text{g mL}^{-1}$	-	Ni(II)-imprinted amino-functionalized silica gel	280.03 Ni over Co selectivity coefficient ^d	6
Adsorption	[Co]: 1 $\mu\text{g mL}^{-1}$ [Ni]: 1 $\mu\text{g mL}^{-1}$	-	Ni(II) ion-imprinted polymer	14.1 Ni over Co selectivity coefficient ^d	7
Intercalation electrode membrane	[Co]: 0.1 M [Ni]: 0.1 M	-	Mo ₆ S ₈ (Chevrel phase) electrochemical transfer junction	99% Co over Ni selectivity factor ^e	8
Electrodialysis	[Co]: 0.01 M [Ni]: 0.01 M	3–6 M HCl solution	Liquid membrane (trialkylbenzylammonium chloride + tri-n-octylamine in 1,2-dichloroethane)	145 Co over Ni separation factor ^f	9
Electrodeposition	[Co]: 0.1 M [Ni]: 0.1 M	10 M LiCl	Poly(diallyldimethylammonium chloride) on copper	16.73 Co over Ni separation factor ^e	This study

^a Separation factor: $(A/B)_{\text{precipitate}}/(A/B)_{\text{initial solution concentration}}$

^b Separation factor is defined as $D_{\text{Co}}/D_{\text{Ni}}$, where D_{metal} is distribution coefficient of a metal in the extraction process.

^c Separation factor: $(A/B)_{\text{adsorbed or deposited}}/(A/B)_{\text{initial solution concentration}}$

^d Selectivity coefficient: $(D_{\text{Ni}}/D_{\text{Co}})$, where $D=Q/C_e$ (Q : adsorption capacity in mg g^{-1} , C_e : equilibrium concentration)

^e Selectivity factor: ratio $n(\text{Co})/(n(\text{Co})+ n(\text{Ni}))$, where n is the number of moles in the recovery compartment.

^f Separation factor: $(A/B)_{\text{in strip solution}}/(A/B)_{\text{in feed solution}}$

The methodology (involving ICP and XPS) and data analysis of the manuscript is detailed and support sufficiently the claims, but I could not find SEM images of the electrode before and after polymer coating in order to judge whether the polymer coating is uniform. The authors should add this comparison based on SEM and check whether there are cracks in the polymer coating on the surface. Could this be a reason for not achieving complete Co (near 100%) separation?

We thank the reviewer for pointing this out. We carried out additional SEM analysis to see whether there are cracks in the polymer film, and the results are added in **Figure S10 and S11**, with corresponding explanation in the manuscript. We found that PDADMA loaded on a pristine copper in general exhibited smooth and uniform coating, but there were unevenly distributed cracks at a relatively higher PDADMA loading (0.75 mg cm^{-2}). We hypothesize that copper surface exposed by this crack can be a conductive area where high ion/electron flux is concentrated, contributing to decreasing selectivity. In the future, we will carry out a follow-up study for uniform polymer coating and impact on selectivity.

This is a manuscript with important results for Co recovery because it presents an energy-efficient and scalable method, but the weak point is the lack of comparison to literature reports to give an overview on the current state of the art in the field. This requires that the manuscript undergoes a major revision before it could be accepted in Nat. Comm. I would be happy to review the revised version of the manuscript.

Again, we thank the reviewer for recognizing the importance of our results. The reviewer's comment on the lack of comparison to literature is addressed above.

Reviewer #2 (Remarks to the Author):

This paper intends to achieve selectivity for cobalt and nickel with the synergistic combination of electrolyte control and interfacial design in the potential-dependent electrodeposition. The chemical properties of nickel and cobalt are similar, prompting the industry to use cheap nickel to replace part of the cobalt in lithium-ion batteries (LIBs), which makes it difficult to achieve a high-level recycling of nickel and cobalt waste in spent LIBs, simultaneously. This article focuses on the separation of nickel and cobalt, which is very interesting, but there are still some problems in the research itself. I will consider his official publication only after a major revise.

1. The concentration of Co and Ni in the simulated solution is only 10 mM, it seems this concentration is much lower than the real situation in presently recovery process. It would be better to add more explanation about the representativeness of the solution.

We thank the reviewer for the careful consideration of manuscript and helpful comments. We targeted lower amounts of Co/Ni as the more challenging process first, as being able to separate cobalt from nickel at low concentrations gives this technique a broader applicability for various leaching processes, but as mentioned below, we can also target higher concentrations. Leaching processes usually benefit from higher liquid:solid ratio, if that gives higher leaching efficiency, not being limited by the subsequent separation step. This is of particular importance when using greener lixivants (*e.g.*, organic acid and deep eutectic solvent with concentrated chloride), which require higher liquid:solid ratios compared to typical inorganic acids, creating dilute streams. All these necessitate the consideration of cobalt/nickel deposition not only in concentrated streams but also diluted streams under disadvantageous boundary conditions.

We also considered how dilute concentration affects electrodeposition performance – electrodeposition at a two-dimensional cathode is most suitable for relatively high initial concentration; therefore we considered demonstration of concept at lower concentrations to be the more challenging task, due to possibility of mass-transfer-control.¹⁰ As shown in **Figure S1**, having larger concentration gives better Co/Ni separation factor, showing our applicability for a range of Co/Ni concentrations. We added explanation about the use of diluted concentrations in *Methods* section.

2. According to the Fig 2(f), it indicated that a higher Co/Ni ratio could be achieved when the potential is -0.75 V, but in the following text and experiment, the potential of 0.725 was used, what's the reason for this difference? Meanwhile, in the Fig 2 e and f, the results should be present as line chart rather than a smooth curve.

We thank the reviewer for pointing this out. Even though -0.75 V leads to a higher Co/Ni ratio according to Figure 2(f), the optimal potential at which high Co/Ni ratio is observed is affected by initial Co/Ni concentration. For example, -0.75 V gives lower Co/Ni ratio (<10) compared to -0.725 V (>14) at 100 mM Co/Ni solution. Thus, in our potential-controlled electrodeposition, in order to have acceptable separation factor in a wide range of concentration, we used -0.725 V. As an effort to handle the reviewer's comment, we changed Figure 2(d)-(f) into a straight line chart, instead of a smooth curve.

3. In Fig. 3A, when the potential is -0.725V, if the PDADMA loading is higher than 5 mg/cm², can selectivity be further improved?

We found that there was slight increase in Ni selectivity at a loading higher than 5 mg cm⁻², but the improvement was not significant. For example, the Co/Ni ratio was 0.40 for the electrode with PDADMA loading of 4.995 mg cm⁻², and it decreased to 0.32 at 15 mg cm⁻². It is worth noting that a loading larger than > 5 mg cm⁻² led to a greatly thick film, with outer layer peeling off easily. For this reason, we carried out our experiment up to a loading of 5 mg cm⁻².

4. The results of the first half of the paper are the experiments carried when Co and Ni are in the same concentration. But when the electro deposition is carried out following the proposed chart, will the different concentrations of the two elements have a significant impact on the recycling performance?

Separation factor, which can be defined as $(\text{Co/Ni})_{\text{deposited}}/(\text{Co/Ni})_{\text{initial solution concentration}}$ can be a metric to evaluate the recycling performance at various initial concentration ratios. In the revised experiment using the spent LIBs after leaching (**Figure 5**), the Co/Ni ratio of our post-leaching electrolyte was 1:6.52, which was significantly lower than the simulated solution. Even so, selective cobalt deposition allowed for the up-concentration of cobalt, resulting in Co/Ni ratio of 1:0.6 after stripping, which translates to a separation factor of about 11 – a similar separation factor compared to equimolar binary solution. The stripped solution can be treated with secondary cobalt deposition unit for further purification. This result indicates that selectivity performance does not significantly deteriorate with the different concentrations, and serial electrodeposition/stripping enables high enrichment factor of cobalt, as demonstrated in our experiments (>96% purity starting from 12.4% purity).

5. In Fig. 5, Why not electrodeposition at 0.6V first and then electrodeposition at 0.725? What is the voltage control sensitivity that can be realized in industrial practice? Is it operational?

The strength of our system is that the operation is flexible depending on inlet concentrations in industrial practice. In nickel-rich stream, on the one hand, one can choose to electrodeposit nickel first at -0.6 V to remove nickel first, followed by cobalt deposition. On the other hand, another option is to electrodeposit at -0.725 V, which allows for selective removal of cobalt with larger deposition capacity (due to larger overpotential and current), even when the concentration of cobalt is lower than nickel – this is contrary to the prediction from thermodynamic Nernst equation and reduction potential, but this happened due to our synergistic electrolyte and interface control. As shown in the revised **Figure 5**, which employed the practical LIBs, we demonstrated that the first selective cobalt electrodeposition helped i) up-concentration of cobalt by deposition/stripping cycle and ii) generation of cobalt-removed stream for subsequent nickel deposition.

Furthermore, as proposed in the revised **Figure 5**, the two electrodeposition units (cobalt-selective and nickel-selective) can work in a complementary way to control Co/Ni ratio in the effluent streams. For example, if the Co/Ni ratio decreases after selective cobalt deposition, the stream can be sent for selective nickel deposition to recover nickel. In a similar way, if Co/Ni ratio increases too high after selective nickel recovery, the stream can be transferred to cobalt-recovering unit. In this revised manuscript, we added the explanation of operation in the *Result* section.

6. Whether the technology can be applied in practice depends on its environmental hazards and economic value. The author must supplement the LCA-based environmental impact assessment and economic assessment of this technology, so as to prove that the technology has published value.

As an effort to incorporate the reviewer's comment, we carried out technoeconomic analysis using materials/energy flow in our lab-scale recovery of cobalt and nickel from the spent end-of-life LIBs. As shown in the revised **Figure 5**, we carefully disassembled and harvested cathode materials from practical 18650 LIBs, and all our calculations were based on the selectivity performance under real-world condition. The technoeconomic analysis was carried out at our experimental scales (e.g., electrolyte volume, working area of the electrode) for a given amount of spent LIBs cathode powder (which was 4 g in our study), then the cost and revenue were normalized to per kg basis. Even though this technique seems to be easily scalable, the use of a batch experimental scale allows for us to be conservative in calculating the energy/material consumptions. The market prices of various industrial-grade reagents were obtained from research papers, reports, and websites. Based on our experimental results during harvesting and sampling from spent LIBs, we found a final revenue of \$0.2 per 1 kg of NMC powder in the whole process. This analysis was carried out under some conservative assumptions of lab-scale 2-D electrodes and unoptimized operations, and we expect further improvement by rational design of electrochemical interfaces, developing better polymer coating, and scaling-up and optimizations in larger scale operations. We believe these initial studies point to the promising scalability and avenue for further studies going forward.

We did not include a full LCA, which we believe is too preliminary and outside the scope or focus of this particular work. Here, the main focus of this present study is to present the fundamental science of electrolyte engineering for speciation control, understand how interfacial charge control affects molecular interaction and selectivity tuning, and envision how this synergistic control can benefit in real life applications using practical spent LIBs along with technoeconomic analysis. Our thorough separation test with practical LIBs, electrochemical characterizations, spectroscopic and electrogravimetric analysis, and technoeconomic analysis all led us to believe that the feasibility and novelty of our system is sufficiently proved through this study. Full LCA, which is also of prime importance and yet sufficiently challenging, could be pursued at a future study.

Reviewer #3 (Remarks to the Author):

This paper proposes the use of electrodeposition to separate nickel from cobalt in waste streams produced from lithium ion batteries. In principle some of the science is passable although I was not surprised by the results. There is no clear urgency to publish these results. The authors unfortunately display a lack of knowledge about lithium ion battery recycling. From the perspective of kinetics, cost, digestion and efficiency this methodology is totally impractical. The work could be publishable in a much lower impact journal if discussion of lithium ion battery recycling was removed.

This work should not be published in any Nature journal

We thank the reviewer for time and consideration of our manuscript. To the best of our knowledge, there has not been any successful selective electrodeposition/separation of cobalt from nickel in isothermal conditions, which is a core problem for enabling electrochemical recycling. However, we acknowledge the importance of practical perspective. As an effort to seriously handle the reviewer's comment, we provided new experimental results in the resubmission – including methodology and analysis using practical NMC cathodes harvested from spent LIBs. In the *Method* section, we carefully reported the method of cell pretreatment – including discharging, dismantling, treatment with organic solvents, and leaching. As added in our *Results and Discussion* section, we demonstrated high purity of cobalt (>96%) and nickel (>94%) recovered using our electrochemical recovery techniques starting from the harvested practical NMC cathodes, proving the capability of our method to effectively recover critical elements.

Notably, in this resubmission, we proved the possibility of anodically stripping the electrodeposited cobalt and nickel. Anodic stripping in a weak acid provided an easy way of releasing recovered solid-phase cobalt/nickel into a liquid phase for secondary up-concentration/processing. Our findings revealed that there is no need to digest the deposits, and proved high regeneration efficiency (>90%) using ICP measurement and negligible polymer loss during the stripping using EQCM analysis, shedding light on the practical applicability of our proposed system for battery recycling. We believe our new experimental results significantly strengthen the technical quality of the paper.

References

1. Yang X, Zhang Y, Meng Q, Dong P, Ning P, Li Q. Recovery of valuable metals from mixed spent lithium-ion batteries by multi-step directional precipitation. *RSC Advances* **11**, 268-277 (2021).
2. Othman EA, van der Ham AGJ, Miedema H, Kersten SRA. Recovery of metals from spent lithium-ion batteries using ionic liquid [P8888][Oleate]. *Separation and Purification Technology* **252**, 117435 (2020).
3. Kang J, Senanayake G, Sohn J, Shin SM. Recovery of cobalt sulfate from spent lithium ion batteries by reductive leaching and solvent extraction with Cyanex 272. *Hydrometallurgy* **100**, 168-171 (2010).

4. Gomaa H, Shenashen MA, Yamaguchi H, Alamoudi AS, El-Safty SA. Extraction and recovery of Co²⁺ ions from spent lithium-ion batteries using hierarchical mesosponge γ -Al₂O₃ monolith extractors. *Green Chemistry* **20**, 1841-1857 (2018).
5. Gomaa H, *et al.* Three-Dimensional, Vertical Platelets of ZnO Carriers for Selective Extraction of Cobalt Ions from Waste Printed Circuit Boards. *ACS Sustainable Chemistry & Engineering* **6**, 13813-13825 (2018).
6. Jiang N, Chang X, Zheng H, He Q, Hu Z. Selective solid-phase extraction of nickel(II) using a surface-imprinted silica gel sorbent. *Analytica Chimica Acta* **577**, 225-231 (2006).
7. Abbasi S, Roushani M, Khani H, Sahraei R, Mansouri G. Synthesis and application of ion-imprinted polymer nanoparticles for the determination of nickel ions. *Spectrochimica Acta Part A: Molecular and Biomolecular Spectroscopy* **140**, 534-543 (2015).
8. Guyot E, *et al.* Mo₆S₈ Electrochemical Transfer Junction for Selective Extraction of Co²⁺-Ions from Their Mixture with Ni²⁺-Ions. *Journal of The Electrochemical Society* **160**, A420-A425 (2013).
9. Sadyrbaeva TZ. Separation of cobalt(II) from nickel(II) by a hybrid liquid membrane–electrodialysis process using anion exchange carriers. *Desalination* **365**, 167-175 (2015).
10. Khazi I, Mescheder U. Micromechanical properties of anomalously electrodeposited nanocrystalline Nickel-Cobalt alloys: a review. *Materials Research Express* **6**, 082001 (2019).
11. Tran MK, Rodrigues M-TF, Kato K, Babu G, Ajayan PM. Deep eutectic solvents for cathode recycling of Li-ion batteries. *Nature Energy* **4**, 339-345 (2019).

EVIEWERS' COMMENTS

Reviewer #1 (Remarks to the Author):

I was happy to read the revised version of the improved manuscript. As a researcher who is working on Li-ion batteries recycling in collaboration with industrial partners on that topic, I consider these results as interesting from both the scientific and technological perspective; similar to my comments in the first round of review.

In the revised version, the authors have addressed all issues I pointed out to a satisfactory level. I have one minor point to address: the new table for comparing the recycling approaches is almost complete. A recent report on Ni-Co separation using green chemistry methodology at room temperature on a MOF is missing (ACS Sustainable Chem. Eng. 2021, 9, 29, 9770–9778). Since the paper is from model solutions, it does not decrease the novelty of the manuscript of Kim. If the table is updated then the manuscript can be accepted without further review.

We thank the reviewer and the editorial office for the valuable suggestions. We have addressed the reviewer comment as seen below, as well as have re-checked the whole manuscript for typos and other grammatical mistakes.

Reviewer #1 (Remarks to the Author):

I was happy to read the revised version of the improved manuscript. As a researcher who is working on Li-ion batteries recycling in collaboration with industrial partners on that topic, I consider these results as interesting from both the scientific and technological perspective; similar to my comments in the first round of review.

In the revised version, the authors have addressed all issues I pointed out to a satisfactory level. I have one minor point to address: the new table for comparing the recycling approaches is almost complete. A recent report on Ni-Co separation using green chemistry methodology at room temperature on a MOF is missing (*ACS Sustainable Chem. Eng.* 2021, 9, 29, 9770–9778). Since the paper is from model solutions, it does not decrease the novelty of the manuscript of Kim. If the table is updated then the manuscript can be accepted without further review.

[Response]

We thank the reviewer for the positive feedback, and support to our work. As the reviewer suggested, we updated the **Supplementary Table 1** and cited the article (*ACS Sustainable Chem. Eng.* 2021, 9, 29, 9770–9778) in the main text. Unfortunately, the paper did not present any numerical value of selectivity performance – the article shows only change in UV-vis spectra after nickel recovery from cobalt/nickel mixture without quantification of both metals. Therefore, we could not put the selectivity metric in **Supplementary Table 1** but provided key materials and solution composition used in the study.